# Mediation Effect of Platelet Traits on Associations of Central Obesity with Aging Biomarkers in Rural Adults of Henan, China

**DOI:** 10.3390/nu14173597

**Published:** 2022-08-31

**Authors:** Xinwei Chen, Ruiying Li, Xiaoyu Hou, Yuqin Wang, Mingming Pan, Ning Kang, Yinghao Yuchi, Wei Liao, Xiaotian Liu, Zhenxing Mao, Wenqian Huo, Chongjian Wang, Jian Hou

**Affiliations:** 1Department of Epidemiology and Biostatistics, College of Public Health, Zhengzhou University, Zhengzhou 450001, China; 2Guangzhou First People’s Hospital, School of Medicine, South China University of Technology, Guangzhou 510006, China

**Keywords:** obesity, telomere length, mitochondrial DNA copy number, platelet traits, mediating effect

## Abstract

Background: To assess the associations of platelet traits and obesity indices with aging biomarkers (telomere length (TL) and mitochondrial DNA copy number (mtDNA-CN)). Methods: A cross-sectional study was performed among 5091 rural Chinese adults. Obesity indices (waist circumference (WC), waist-to-hip ratio (WHR) and waist-to-height ratio (WHtR)) and platelet traits (plateletcrit (PCT), platelet large cell ratio (P-LCR), mean platelet volume (MPV) and platelet distribution width (PDW)) were collected by physical examination. The aging biomarkers were determined by quantitative real-time polymerase chain reaction. Generalized linear regression models and mediation analysis were applied to explore the relationships between platelet traits, obesity indices, and aging biomarkers. Results: The mean age of the participants was 56 years (range: 18–79). Each one-unit increment in WC, WHR and WHtR were related to a 0.316 (95% confidence interval (CI): −0.437, −0.196), 0.323 (95% CI: −0.513, −0.134) and 0.277 (95% CI: −0.400, −0.153) decrease in relative TL; or a 0.102 (95% CI: −0.197, −0.007), 0.109 (95% CI: −0.258, −0.041) and 0.101 (95% CI: −0.199, −0.004) decrease in relative mtDNA-CN. The proportions of obesity indices with aging biomarkers mediated by platelet indices ranged from 2.85% to 10.10%. Conclusions: Increased central obesity indices in relation to shortened relative TL or decreased mtDNA-CN were mediated by platelet traits, indicating that obesity in association with the accelerated aging process may be partially attributable to abnormal platelet activity.

## 1. Introduction

Both telomere length (TL) and mitochondrial dysfunction have been accepted as the molecular biomarkers of aging [1,2]. Telomeres are nucleoprotein structures that maintain genome stability by serving as caps on the ends of chromosomes [3]. Telomere attrition is commonly considered to reflect the replicative capacity, and TL is shortened with each cell division [3]. Cell aging can be induced by telomere uncapping via oxidative stress, indicating the measurement of telomere length as a biomarker of chronic oxidative stress [4]. Meanwhile, a bulk of studies have also shown that decreased mitochondrial DNA copy number (mtDNA-CN), as a good biomarker to reflect the content and function of mitochondria, can be induced by oxidative stress response and ultimately lead to a series of adverse events [5,6,7]. For instance, a study focused on Danish twins and singletons indicated that those with high mtDNA-CN in blood were associated with better health and survival [8]. Both shortened TL and decreased mtDNA-CN are related to age-related diseases such as cardiovascular disease, cancer and diabetes [9,10,11,12].

Obesity has become a huge public health problem. Several meta-analyses have shown a negative association between obesity and TL [13,14]. A cross-sectional sample of 309 non-Hispanic white participants aged 8 to 80 years in the Fels Longitudinal Study indicated that individuals with higher total and abdominal adiposity had shortened TL [15]. Another study has even shown that decline in newborn TL occurred in parallel with higher maternal pre-pregnancy body mass index (BMI) [16]. Obesity also has adverse effects on mitochondrial function, which may lead to a decrease in mtDNA-CN [17,18]. Munusamy et al. indicated that reduction in mitochondria-related decreased cellular ATP levels may be involved in obesity-induced renal injury [18]. A study showed increased mtDNA-CN in peripheral blood obtained from 46 obese patients with type 2 diabetes after bariatric surgery for 1 year [19].

Obesity-related altered platelet activity has been widely reported [20,21,22]. Obesity is known to activate platelets and affect their size and function [22,23,24]. For instance, a cohort study of 17,327 participants showed that participants with a higher BMI had a greater risk of low mean platelet volume (MPV) and platelet distribution width (PDW), and this was more significant in subjects with higher waist-to-height ratio (WHtR) and waist circumstance (WC) [22]. Erdal and Inanir found that morbid obesity was associated with increased platelet counts and plateletcrit (PCT) values [24]. Platelets are produced by megakaryocytes and play a major role in hemostasis and thrombosis regulation [25]. Moreover, platelets can be involved in immune response and inflammatory response, and subsequently contribute to the process of aging [26,27]. However, the evidence on the role of platelet activity in the association of obesity with aging was rare in low-middle income countries; thus, a cross-sectional study was conducted in Henan rural adults to examine the role of platelet activity reflected by PCT, platelet large cell ratio (P-LCR), MPV and PDW in associations between obesity and aging biomarkers.

## 2. Material and Methods

### 2.1. Study Population

This study’s participants were derived from the sub-population (N = 6670) of Henan Rural Cohort (N = 39,259), which has been described elsewhere [28]. After participants with missing data on demographic characteristics (N = 205), obesity indices (WC, waist-to-hip ratio (WHR), and WHtR) (N = 11), platelet traits (PCT, P-LCR, MPV, and PDW) (N = 1322), aging biomarkers (TL and mtDNA-CN) (N = 29) as well as pregnant women (N = 12) were excluded, a total of 5091 participants were used for this study analysis. This study was conducted with the approval of Zhengzhou University Life Science Ethics Committee (Ethic approval code: [2015] MEC (S128)), and written informed consent was obtained from each participant.

### 2.2. Data Collection

A standardized questionnaire was performed to collect individuals’ information on demographic characteristics (sex, age, average monthly income, educational level, marital status, etc.), lifestyle behaviors (smoking status, alcohol consumption status, physical activity, dietary habits, etc.) and histories of diseases (hypertension, hyperlipidemia, diabetes, etc.). Both smoking and drinking status were classified into three groups: never, former and current. Educational level was classified into elementary school or below, junior high school, as well as high school or above groups. Physical activity was grouped into low, moderate and high levels, as mentioned in our previous study [29].

### 2.3. Measurement of Obesity Indices

Individuals’ physical examinations (such as body height, weight, and WC) were measured in accordance with the standard operating instructions. Body height and weight were measured twice under the condition of individual with light clothing with shoes off. WC and hip circumference were measured twice at 1 cm above the navel and the maximal level of the hip using a standard flexible rule, respectively. The WHR values were calculated as WC divided by hip circumference. The WHtR values were calculated as waist circumference divided by body height.

### 2.4. Measurement of Platelet Traits

After overnight fasting for at least 8 h, venous blood samples were obtained from each participant. A Sysmex XT−500i automated hematology analyzer (Sysmex Corporation, Kobe, Japan) was used to measure platelet parameters including MPV, PDW, P-LCR and PCT.

### 2.5. Measurement of Aging Biomarkers

Whole genomic DNA of each participant was extracted from peripheral-blood samples by using the whole blood genomic DNA medium extraction kit III (Bioteke Corporation, Beijing, China) in accordance with the manufacturer’s protocols. Real-time polymerase chain reaction (RT-PCR) (QuantStudio™ 7 Flex, AppliedBiosystems Life technologies, Waltham, MA, USA) was used to measure the telomere repeat copy number, mitochondrial DNA copy number, as well as single copy gene copy number in peripheral leukocytes in triplicate [30]. The forward and reverse primers of these genes have been previously reported elsewhere [29,31,32]. The relative mtDNA-CN or TL values were calculated by the average of three measurements of mtDNA or the telomere repeat copy number divided by the average of three measurements of single copy gene copy number using the 2^−ΔΔCt^ method [33].

### 2.6. Statistical Analysis

The non-normal distributed continuous variables and categorical variables were expressed as median (Q25, Q75) and number (percentage), respectively. Obesity indices (WC, WHR, and WHtR), platelet traits (PCT, P-LCR, MPV, and PDW), and aging biomarkers (TL and mtDNA-CN) were natural logarithmic transformed to approximate normal distribution before the main analysis. Spearman rank correlation was performed to evaluate the pairwise correlation between obesity indices and platelet traits. Generalized linear regression models were applied to evaluate the independent associations of obesity indices with aging biomarkers or platelet traits and the relationship between obesity indices and platelet traits. Three models were built: Model 1 was unadjusted; Model 2 was adjusted for individuals’ demographic characteristics (such as age, sex, personal average monthly income, marital status, and educational attainment) and lifestyle behaviors (such as physical activity, drinking and smoking status, vegetable and fruit intake, and high fat diet); Model 3 was additionally adjusted for hypertension, type 2 diabetes mellitus, dyslipidemia, and family histories of hypertension, hyperlipidemia, and diabetes. Furthermore, mediation analysis was applied to evaluate the role of platelet traits in the associations between obesity indices and aging biomarkers (as shown in Figure 1). All analyses in this study were conducted using R software (version 3.5.1, R foundation for Statistical Computing, Vienna, Austria), and *p* values (two-tailed) of less than 0.05 were defined as statistically significant.

## 3. Results

### 3.1. Characteristics of Study Participants

Table 1 shows that the median age was 55.0 years; the proportion of men, elementary school or below, current-smokers, current-drinkers, middle-high physical activity, high-fat diet intake, and more vegetable and fruit intake was 42.0%, 47.5%, 20.5%, 15.6%, 50.2%, 27.1%, and 66.6%, respectively; the proportions of type 2 diabetes mellitus (T2DM), hypertension, and hyperlipidemia were 5.3%, 23.6%, and 33.0%, respectively. Table 2 shows that the medians (IQR) of WC, WHR and WHtR were 82.80 cm (13.70 cm), 0.88 (0.09), and 0.52 (0.09), respectively; the medians (IQR) of PCT, P-LCR, MPV and PDW were 0.23 (0.08), 41.80% (14.80%), 12.10 fl (2.00 fl), and 15.70 fl (5.10 fl), respectively; the medians (IQR) of relative TL and mtDNA-CN were 0.60 (0.49) and 0.78 (0.37), respectively. Figure 2 depicts that obese participants tended to have lower PLCR, MPV, PDW values and higher PCT values relative to the non-obese ones.

### 3.2. Association of Obesity Indices with Aging Biomarkers

As shown in Figure 3, each 1-unit increase in WC, WHR or WHtR value was associated with a 0.316 (95% CI: −0.437, −0.196), 0.323 (95% CI: −0.513, −0.134) or 0.277 (95% CI: −0.400, −0.153) decrease in relative TL value as well as a 0.102 (95% CI: −0.197, −0.007), 0.109 (95% CI: −0.258, 0.041) or 0.101 (95% CI: −0.199, −0.004) decrease in mtDNA-CN value (Model 3).

### 3.3. Association of Platelet Traits with Aging Biomarkers

As shown in Figure 4, each 1-unit increase in PLCR, MPV or PDW value was associated with a 0.097 (95% CI: 0.043, 0.152), 0.260 (95% CI: 0.134, 0.386) or 0.135 (95% CI: 0.071, 0.199) increase in relative TL values as well as a 0.070 (95% CI: 0.027, 0.112), 0.182 (95% CI: 0.083, 0.282) or 0.098 (95% CI: 0.048, 0.149) increase in mtDNA-CN value; each 1-unit increase in PCT value was associated with a 0.074 (95% CI: −0.124, −0.025) decrease in relative TL value and a 0.081 (95% CI: 0.042, 0.120) increase in mtDNA-CN value (Model 3).

### 3.4. Association of Obesity Indices with Platelet Traits

As shown in the Figure 5, each 1-unit increase in WC, WHR or WHtR was associated with a 0.156 (95% CI: 0.089, 0.223), 0.249 (95% CI: 0.144, 0.354) or 0.178 (95% CI: 0.109, 0.246) increase in PCT values; a 0.110 (95% CI: −0.171, −0.049), 0.172 (95% CI: −0.268, −0.077) or 0.127 (95% CI: −0.189, −0.064) decrease in PLCR values; a 0.049 (95% CI: −0.075, −0.023), 0.074 (95% CI: −0.115, −0.033) or 0.055 (95% CI: −0.082, −0.028) decrease in MPV values; as well as a 0.065 (95% CI: −0.117, −0.013), 0.100 (95% CI: −0.181, −0.019) or 0.076 (95% CI: −0.129, −0.023) decrease in PDW values (Model 3).

### 3.5. Mediating Role of Platelet Traits in Associations of Obesity Indices with Aging Biomarkers

Table 3 shows that platelet traits are potential mediators between obesity indices and aging biomarkers. The proportion of mediation effect of PCT, P-LCR, MPV or PDW on the associations between WC and relative TL was 3.48%, 3.16%, 3.80% or 2.85%, respectively; the proportion of mediation effect of PCT, P-LCR, MPV or PDW on the associations between WHR and relative TL was 5.56%, 4.94%, 5.86% or 4.01%, respectively; the proportion of mediation effect of PCT, P-LCR, MPV or PDW on the associations between WHtR and relative TL was 4.64%, 4.29%, 5.00% or 3.57%, respectively. The proportion of mediation effect of P-LCR, MPV or PDW on the associations between WC and mtDNA-CN was 8.25%, 9.28% or 6.19%, respectively; the proportion of mediation effect of P-LCR, MPV or PDW on the associations between WHtR and mtDNA-CN was 9.09%, 10.10% or 8.08%, respectively.

## 4. Discussion

Central obesity indices (WC, WHR, and WHtR) were positively associated with accelerated aging reflected by shortened TL and decreased mtDNA-CN, and these associations were mediated by platelet traits, implying that platelet dysfunction may be a potential biological mechanism of central obesity-related aging.

Central obesity in relation to TL shortening was in line with several previous studies. A cross-sectional study conducted in 479 Paris participants indicated that each 10 cm increased in WC was related to a 1.30-fold increase in TL shortening (95% CI: 1.11, 1.52) [34]. Another study conducted in 768 mother–newborn pairs in Limburg, Belgium reported that the pre-pregnancy body mass index of pregnant women was negatively related to TL in newborns, which was measured by using umbilical cord blood and placenta specimens [16]. However, some studies failed to observe the association between obesity and TL [35,36]. Additionally, limited evidence showed that peripheral blood leucocyte mtDNA-CN was negative associated with some obesity parameters such as WC or body mass index among 14,176 participants in Germany and Italy [17]. Although the mechanisms of the obesity-related aging process were unclear, evidence indicated that obesity may lead to the dysfunction of telomeres and mitochondria, inducing oxidative stress. A previous study showed that the excessive nutrient supply, as a risk factor for obesity, may induce excessive reactive oxygen species (ROS) generation by modulating the Krebs cycle and the mitochondrial respiratory chain and ultimately lead to mitochondrial dysfunction [37]. Accumulated evidence suggested that oxidative stress is linked to shortened TL [13,38]. An experimental study showed that chronic inflammation aggravated telomere dysfunction and accelerated aging in NF-κB^−/−^ mice and TL, also in association with inflammatory markers such as TNFα and C-reactive protein, which suggested the inflammation involved in telomere shortening [39]. Furthermore, obesity may be related to nervous system, kidney, and blood vessel function disorders by inducing mitochondrial dysfunction [40,41,42]. For instance, evidence suggested that obese individuals with decreased mtDNA-CN were at a high risk for increased TC and prevalent type 2 diabetes mellitus [17,43].

A negative association of PCT with TL was consistent with a previously reported study that suggested a negative correlation of relative TL with platelet count among German females (*r* = −0.091, *p* = 0.015) [44]. This study also indicated positive associations of platelet traits (P-LCR, MPV, and PDW) with aging markers (mtDNA-CN). Evidence suggested that platelets may release large amounts of inflammatory mediators that recruit white blood cells from blood vessels into tissues to perform inflammatory functions, and ultimately lead to altered TL and mitochondrial function with a ROS-dependent pathway [39,45,46]. One study indicated that TL in immune thrombocytopenia patients was shorter relative to that in healthy controls [47]. However, limited studies failed to observe associations between platelet traits and aging biomarkers [48,49] or report the opposite results [36,47].

Moreover, the associations of obesity indices (WC, WHR, and WHtR) with aging biomarkers (relative TL and mtDNA-CN) may be partially attributed to platelet traits (PCT, PLCR, MPV, and PDW). Previous studies suggested the potential mechanisms as follows: central obesity may be linked to platelet dysfunction by inducing the alteration of intracellular ionic milieu, decreasing sensitivity to insulin and other substances that function through intracellular cyclic nucleotides, and increasing oxidative stress, and finally contributing to accelerated aging reflected by telomere and mitochondrial dysfunction [50,51,52,53]. One study indicated that platelet dysfunction in obese females was related to decreased platelet SERCA3 expression, while obese patients with increased platelet SERCA3 expression lost weight, compared to lean control females [54]. Additionally, a genome-wide analysis revealed that platelet activation-related SNPs were related to mtDNA-CN contents among 465,809 white individuals [55]. As mentioned above, we may conclude that the results of this study are plausible.

Evidence revealed that mtDNA-CN and TL, as early molecular biomarkers of aging, may lead to the occurrence of age-related diseases such as diabetes and cardiovascular disease by inducing oxidative stress or inflammatory response. Identifying the risk factors of early aging molecular biomarkers plays an important role in preventing age-related diseases. In the present study, we found that increased obesity indices (WC, WHR, WHtR) were associated with accelerated aging reflected by shortened relative TL and decreased mtDNA-CN, and these associations were partially mediated by platelet traits, implying that obesity-related platelet dysfunction may be one potential biological pathway of the aging process, which suggests that maintaining normal body weight and platelet function monitoring may be useful to prevent aging-related diseases.

## 5. Limitations

Several limitations of this study need to be noted. First, since this was a cross-sectional study, the causal associations could not be evaluated. Second, although several important confounders were controlled, many other uncontrolled factors such as chronic psychological stress and genetic factors in relation to outcome were not considered. Third, this study’s population was selected from one province in China, which might limit generalization to other rural regions. Therefore, more studies are needed to confirm the results of this study in rural populations.

## Figures and Tables

**Figure 1 nutrients-14-03597-f001:**
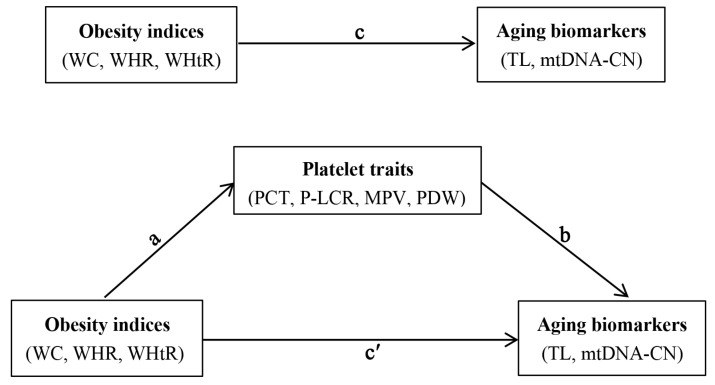
Conceptual mediation model of the associations between obesity indices, platelet traits and aging biomarkers. WC: waist circumference; WHR: waist-to-hip ratio; WHtR: waist-to-height ratio; PCT: plateletcrit; P-LCR: platelet large cell ratio; MPV: mean platelet volume; PDW: platelet distribution width; TL: telomere length; mtDNA-CN: mitochondrial DNA copy number. Path a indicates the effect of obesity indices (WC, WHR, and WHtR) on platelet traits (PCT, P-LCR, MPV, and PDW); path b indicates the effect of platelet traits on aging biomarkers (TL and mtDNA-CN); path c reflects the total effect of obesity indices on aging biomarkers; path c′ reflects the effect of obesity indices on aging biomarkers when controlled for platelet traits.

**Figure 2 nutrients-14-03597-f002:**
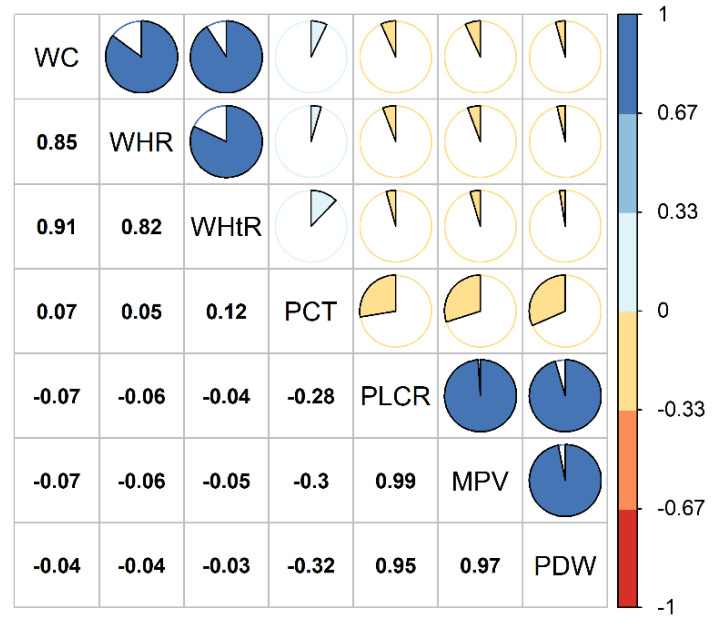
Spearman rank correlation between obesity indices and platelet traits. MPV: mean platelet volume; PCT: plateletcrit; PDW: platelet distribution width; P-LCR: platelet large cell ratio; WC: waist circumference; WHR: waist-to-hip ratio; WHtR: waist-to-height ratio.

**Figure 3 nutrients-14-03597-f003:**
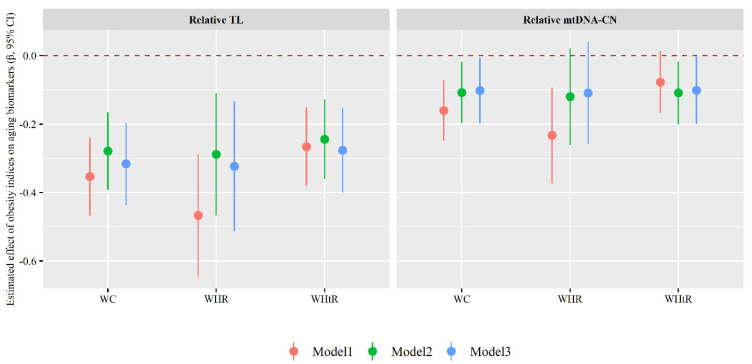
Associations of obesity indices with aging biomarkers. Model 1 was the crude model; Model 2 was adjusted for individuals’ demographic characteristics (such as age, sex, personal average monthly income, marital status, and educational attainment) and lifestyle behavior (such as physical activity, drinking and smoking status, vegetable and fruit intake, and high fat diet); Model 3 was additionally adjusted for hypertension, type 2 diabetes mellitus, dyslipidemia, and family histories of diseases including hypertension, hyperlipidemia, or diabetes. mtDNA-CN: mitochondrial DNA copy number; TL: telomere length; WC: waist circumference; WHR: waist-to-hip ratio; WHtR: waist-to-height ratio.

**Figure 4 nutrients-14-03597-f004:**
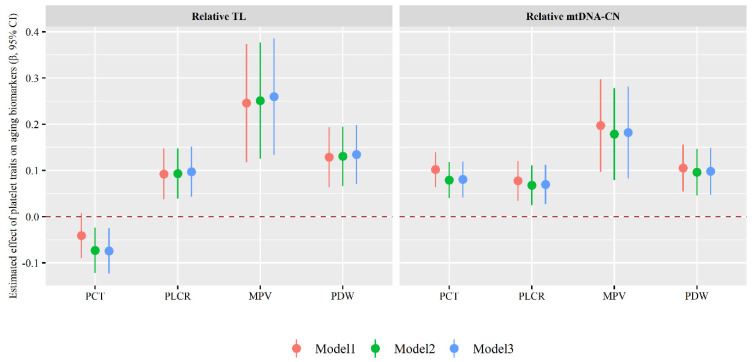
Associations of platelet traits with aging biomarkers. Model 1 was the crude model; Model 2 was adjusted for individuals’ demographic characteristics (such as age, sex, personal average monthly income, marital status, and educational attainment) and lifestyle behavior (such as physical activity, drinking and smoking status, vegetable and fruit intake, and high fat diet); Model 3 was an additional adjustment of hypertension, type 2 diabetes mellitus, dyslipidemia, and family histories of diseases including hypertension, hyperlipidemia, or diabetes. MPV: mean platelet volume; mtDNA-CN: mitochondrial DNA copy number; PCT: plateletcrit; P-LCR: platelet large cell ratio; PDW: platelet distribution width; TL: telomere length.

**Figure 5 nutrients-14-03597-f005:**
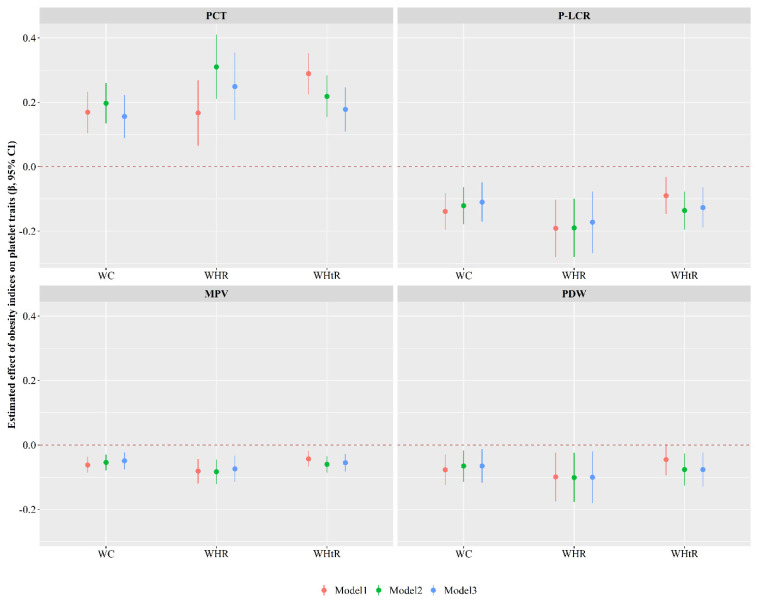
Associations of obesity indices with platelet traits. Model 1 was the crude model; Model 2 was adjusted for individuals’ demographic characteristics (such as age, sex, personal average monthly income, marital status, and educational attainment) and lifestyle behaviors (such as physical activity, drinking and smoking status, vegetable and fruit intake, and high fat diet); Model 3 was additionally adjusted for hypertension, type 2 diabetes mellitus, dyslipidemia, and family histories of diseases including hypertension, hyperlipidemia, and diabetes. MPV: mean platelet volume; PCT: plateletcrit; PDW: platelet distribution width; P-LCR: platelet large cell ratio; WC: waist circumference; WHR: waist-to-hip ratio; WHtR: waist-to-height ratio.

**Table 1 nutrients-14-03597-t001:** Demographic characteristics of all participants.

Characteristic	Median (Q25, Q75) or Number (%)
Age (years) ^a^	55.0 (49.0, 64.0)
Gender (Men) ^b^	2138 (42.0)
Average monthly income (yuan, Chinese RMB) ^b^	
<500	1650 (32.4)
500~999	1549 (30.4)
≥1000	1892 (37.2)
Education level ^b^	
Elementary school or below	2420 (47.5)
Junior high school	2151 (42.3)
High school or above	520 (10.2)
Marital status ^b^	
Married/Cohabitation	4566 (89.7)
Unmarried/divorced/widowed	525 (10.3)
Smoking status ^b^	
Never smokers	3639 (71.5)
Former smokers	410 (8.1)
Current smokers	1042 (20.5)
Alcohol consumption ^b^	
Never drinkers	3971 (78.0)
Former drinkers	325 (6.4)
Current drinkers	795 (15.6)
Physical activity ^b^	
Low	1126 (22.1)
Moderate	2557 (50.2)
High	1408 (27.7)
High fat diet ^b^	1382 (27.1)
More vegetable and fruit intake ^b^	3391 (66.6)
T2DM ^b^	269 (5.3)
Hypertension ^b^	1200 (23.6)
Hyperlipidemia ^b^	1682 (33.0)
Family history of T2DM ^b^	113 (2.2)
Family history of hypertension ^b^	948 (18.6)
Family history of hyperlipidemia ^b^	228 (4.5)

RMB: renminbi (yuan); T2DM: type 2 diabetes mellitus. ^a^ Continuous variables were expressed as median (Q25, Q75); ^b^ Categorical variables were expressed as number (percentage).

**Table 2 nutrients-14-03597-t002:** Distributions of obesity indices, platelet traits, and aging biomarkers.

Variables	Mean	Q25	Median	Q75	IQR
Obesity indices					
WC (cm)	82.81	75.90	82.80	89.60	13.70
WHR (%)	0.88	0.84	0.88	0.93	0.09
WHtR (%)	0.52	0.47	0.52	0.56	0.09
Platelet traits					
PCT (%)	0.23	0.19	0.23	0.27	0.08
P-LCR (%)	42.03	34.60	41.80	49.40	14.80
MPV (fl)	12.20	11.20	12.10	13.20	2.00
PDW (fl)	16.27	13.50	15.70	18.60	5.10
Aging biomarkers					
Relative TL	0.72	0.43	0.60	0.92	0.49
Relative mtDNA-CN	0.85	0.63	0.78	1.00	0.37

IQR: interquartile range; MPV: mean platelet volume; mtDNA-CN: mitochondrial DNA copy number; PCT: plateletcrit; PDW: platelet distribution width; P-LCR: platelet large cell ratio; TL: telomere length; WC: waist circumference; WHR: waist-to-hip ratio; WHtR: waist-to-height ratio.

**Table 3 nutrients-14-03597-t003:** Total and indirect effect of obesity indices with aging biomarkers mediated by platelet traits.

	Relative TL			Relative mtDNA-CN		
	Total Effect (95% CI)	Indirect Effect (95% CI)	Proportion of Mediation (%)	Total Effect (95% CI)	Indirect Effect (95% CI)	Proportion of Mediation (%)
PCT						
WC	−0.316 (−0.437, −0.195)	−0.011 (−0.022, −0.004)	3.48	−0.097 (−0.192, −0.001)	0.013 (0.005, 0.024)	- ^a^
WHR	−0.324 (−0.514, −0.134)	−0.018 (−0.035, −0.007)	5.56	−0.100 (−0.249, 0.050)	0.020 (0.009, 0.037)	- ^a^
WHtR	−0.280 (−0.404, −0.156)	−0.013 (−0.025, −0.005)	4.64	−0.099 (−0.197, −0.001)	0.015 (0.007, 0.026)	- ^a^
P-LCR						
WC	−0.316 (−0.437, −0.195)	−0.010 (−0.021, −0.003)	3.16	−0.097 (−0.192, −0.001)	−0.008 (−0.016, −0.003)	8.25
WHR	−0.324 (−0.514, −0.134)	−0.016 (−0.033, −0.006)	4.94	−0.100 (−0.249, 0.050)	−0.012 (−0.025, −0.004)	- ^a^
WHtR	−0.280 (−0.404, −0.156)	−0.012 (−0.023, −0.004)	4.29	−0.099 (−0.197, −0.001)	−0.009 (−0.018, −0.003)	9.09
MPV						
WC	−0.316 (−0.437, −0.195)	−0.012 (−0.024, −0.005)	3.80	−0.097 (−0.192, −0.001)	−0.009 (−0.018, −0.003)	9.28
WHR	−0.324 (−0.514, −0.134)	−0.019 (−0.036, −0.007)	5.86	−0.100 (−0.249, 0.050)	−0.013 (−0.027, −0.005)	- ^a^
WHtR	−0.280 (−0.404, −0.156)	−0.014 (−0.026, −0.006)	5.00	−0.099 (−0.197, −0.001)	−0.010 (−0.019, −0.004)	10.10
PDW						
WC	−0.316 (−0.437, −0.195)	−0.009 (−0.019, −0.002)	2.85	−0.097 (−0.192, −0.001)	−0.006 (−0.014, −0.002)	6.19
WHR	−0.324 (−0.514, −0.134)	−0.013 (−0.029, −0.004)	4.01	−0.100 (−0.249, 0.050)	−0.010 (−0.022, −0.002)	- ^a^
WHtR	−0.280 (−0.404, −0.156)	−0.010 (−0.021, −0.003)	3.57	−0.099 (−0.197, −0.001)	−0.008 (−0.016, −0.002)	8.08

CI: confidence interval; MPV: mean platelet volume; mtDNA-CN: mitochondrial DNA copy number; PCT: plateletcrit; PDW: platelet distribution width; P-LCR: platelet large cell ratio; TL: telomere length; WC: waist circumference; WHR: waist-to-hip ratio; WHtR: waist-to-height ratio. ^a^ Proportion mediated cannot be calculated.

## Data Availability

The datasets used and/or analyzed during the current study are available from the corresponding author on reasonable request.

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
