# Peer review of "Mediation Effect of Platelet Traits on Associations of Central Obesity with Aging Biomarkers in Rural Adults of Henan, China"

_nutrients, 2022, doi:10.3390/nu14173597_

Round 1

Reviewer 1 Report

Table 1 - It is not clear what value is given in parentheses if the value is one? According to the information at the top of the table, there should be two values - Q25 or Q75?

Explanation of abbreviations under table 2 and figure 2 should be arranged alphabetically - check and correct for all graphics

The study limitations section should be separated from the discussion section

The discussion section lacks a clinical perspective - what is the value of the study? How does it affect a specific population? What can we do with the test results?

The authors themselves admit in their study limitations that only the population of one region was tested - in my opinion, this should translate into the title of the article - instead of rural adults, the population / region studied should be given

Author Response

Dear Editor and reviewers,

 Re: Revise and resubmit a manuscript - “Mediation effect of platelet traits on associations of central obesity with aging biomarkers in rural adults” (No.: nutrients-1821220)

We would like to thank you for giving us a chance to revise the manuscript and the reviewers for giving us the valuable and helpful suggestions concerning our manuscript. Those comments are all valuable and helpful for improving our paper, as well as the important guiding significance to us researches. We have read comments carefully and all modifications are marked in the revised manuscript. A point-by-point response to the reviewer’s comments is listed below. We hope that the revised paper could satisfy you and be accepted for publication.

Many thanks

Yours sincerely,

Dr. Jian Hou

Reviewer 1

1. Table 1 - It is not clear what value is given in parentheses if the value is one? According to the information at the top of the table, there should be two values - Q25 or Q75?

Response: We are sorry for confusing you with the unclear description. In Table 1, continuous variable such as age was represented by median (Q25, Q75), while categorical variables (e.g., gender, average monthly income level, education level, marital status, smoking status, alcohol consumption, physical activity, etc.) were represented by number (percentage). In the revised manuscript, we added the following footnotes into the revised Table 1 for readers to better understand: “a continuous variables were expressed as median (Q25, Q75); b categorical variables were expressed as number (percentage).” (Marked in blue, Page 6).

2. Explanation of abbreviations under table 2 and figure 2 should be arranged alphabetically - check and correct for all graphics.

Response: Thanks for your carefully reading our manuscript. In the revised manuscript, we did.

3. The study limitations section should be separated from the discussion section.

Response: Thank you for constructive and insightful comment. We had separated limitations section from discussion section as a new section (Marked in blue, Page 14, Lines 346-355)

4. The discussion section lacks a clinical perspective - what is the value of the study? How does it affect a specific population? What can we do with the test results?

Response: Thanks for your kind suggestion which give us much help to perfect this manuscript. We added the following sentences “Evidence revealed the mtDNA-CN and TL as the early molecular biomarkers of aging may lead to the occurrence of age-related diseases such as diabetes and cardiovascular disease by inducing oxidative stress or inflammatory response. Identifying the risk factors of early aging molecular biomarkers plays an important role to prevent age-related diseases. In the present study, we found that negative associations between obesity indices (WC, WHR, WHtR) and aging biomarkers (relative TL and mtDNA-CN), and these associations were partially mediated by platelet traits implying that obesity-related platelet dysfunction may be as one potential biological pathway of aging process which suggest that maintaining normal body weight and platelet function monitoring may be useful to prevent aging-related diseases.” in the Discussions section of the revised manuscript. (Marked in blue, Page 14, Lines 334-345)

5. The authors themselves admit in their study limitations that only the population of one region was tested - in my opinion, this should translate into the title of the article - instead of rural adults, the population / region studied should be given.

Response: Thanks for your kind suggestions which give us much help to perfect this manuscript. According to your suggestion, the original title “Mediation effect of platelet traits on associations of central obesity with aging biomarkers in rural adults” was modified as follows: “Mediation effect of platelet traits on associations of central obesity with aging biomarkers in rural adults of Henan, China” in the revised manuscript. (Marked in blue, Page 1, Lines 2-3)

Reviewer 2 Report

Dear Authors,

The manuscript presents an interesting and important topic regarding the relationship between platelet traits and obesity indices with molecular aging among rural Chinese adults. 

The authors concluded that obesity is related to aging, and abnormal platelet parameters are partially involved in this mechanism. The authors performed the study on a large population with proper methodology.

I have some minor considerations regarding missing elements in the manuscript.

  1. Did mitochondrial DNA in this study derive from PBMC cells or serum cell-free mtDNA? Please precise it. (lines 51,66)
  2. Line 73: Did the Authors mean waist to height ratio?
  3. Line 108: Please add the number of the Zhengzhou University Life Science Ethics Committee approval in the material methods section.
  4. Line 188: In my opinion for international readers, please add that remidibi is the other name of Yuan - a formal Chinese currency.
  5. In the abstract, some OR values appear without a minus sign before the value.
  6. Lines 212-220: I hope this text belongs to the description section of Figure 3.
  7. Line 304: Exact values in the discussion section are not necessary.
  8. „Measurement of aging biomarkers” subsection: A substantial part of results derived from molecular biology techniques application; thus, Authors should at least include exact sequences of primers for studied genes and housekeeping genes in the study along with conditions of reactions. It can be provided as supplementary material).
  9. The Authors may consider moving Figure 1 to the material and methods section.

Author Response

Dear Editor and reviewers,

 Re: Revise and resubmit a manuscript - “Mediation effect of platelet traits on associations of central obesity with aging biomarkers in rural adults” (No.: nutrients-1821220)

 We would like to thank you for giving us a chance to revise the manuscript and the reviewers for giving us the valuable and helpful suggestions concerning our manuscript. Those comments are all valuable and helpful for improving our paper, as well as the important guiding significance to us researches. We have read comments carefully and all modifications are marked in the revised manuscript. A point-by-point response to the reviewer’s comments is listed below. We hope that the revised paper could satisfy you and be accepted for publication.

Many thanks

Yours sincerely,

Dr. Jian Hou

Reviewer 2

Q1 Did mitochondrial DNA in this study derive from PBMC cells or serum cell-free mtDNA? Please precise it. (lines 51,66)

Response: We are sorry for confusing you with the unclear description. Telomere length and mitochondrial DNA copy number were detected from peripheral leukocytes. In the revised manuscript, the sentence “The real-time polymerase chain reaction (RT-PCR) (QuantStudio™ 7 Flex, AppliedBiosystems Life technologies, USA) was used to measure the telomere repeat copy number, mitochondrial DNA copy number as well as single copy gene copy number in triplicate.” has been changed as “The real-time polymerase chain reaction (RT-PCR) (QuantStudio™ 7 Flex, AppliedBiosystems Life technologies, USA) was used to measure the telomere repeat copy number, mitochondrial DNA copy number as well as single copy gene copy number in peripheral leukocytes of three times.” (Marked in blue, Page 3, Lines 127-131).

Q2 Line 73: Did the Authors mean waist to height ratio?

Response: Thanks for your kind suggestions which give us much help to perfect this manuscript. We are sorry for this mistake. In the revised manuscript, the words “weight-to-height ratio (WHtR)” have been changed as “waist-to-height ratio (WHtR)” (Marked in blue, Page 2, Line 73)

Q3 Line 108: Please add the number of the Zhengzhou University Life Science Ethics Committee approval in the material methods section.

Response: Thanks for your insightful suggestion. the number of the Zhengzhou University Life Science Ethics Committee approval (Ethic approval code: [2015] MEC (S128)) has been supplemented in the Material and methods section of the revised manuscript: “This study was conducted with the approval of Zhengzhou University Life Science Ethics Committee (Ethic approval code: [2015] MEC (S128)) and the written informed consent was obtained from each participant.” (Marked in blue, Page 3, Lines 94-97)

Q4 Line 188: In my opinion for international readers, please add that remidibi is the other name of Yuan - a formal Chinese currency.

Response: Thanks for your kind suggestions which give us much help to perfect this manuscript. According to your suggestion, “Average monthly income (RMB)” has been changed as “Average monthly income (yuan, Chinese RMB)” in revised Table 1, and the explanation of abbreviation under table 1 “RMB: renminbi” has been changed as “RMB: renminbi (yuan)”. (Marked in blue, Page 6)

Q5 In the abstract, some OR values appear without a minus sign before the value.

Response: Thanks for your insightful suggestion. All the odds ratios in the Abstract section have been checked and revised. (Marked in blue, Page 1)

Q6 Lines 212-220: I hope this text belongs to the description section of Figure 3.

Response: Thanks for your insightful suggestion. In the revised manuscript, we adjusted the orders of the texts to correspond to the Figures.

Q7 Line 304: Exact values in the discussion section are not necessary.

Response: Thanks for your insightful suggestion. According to your suggestion, the exact values in the discussions section were removed. The sentence “Limited evidence showed that the estimated odds ratio (95%CI) of increased WC or body mass index was 1.045 (1.027, 1.064) or 1.017 (1.008, 1.026) in response to each 10-unit decrease in mtDNA-CN among 14,176 participants in Germany and Italy.” has been changed as “Additionally, limited evidence showed that peripheral blood leucocytes mtDNA-CN was negative associated with some obesity parameters such as WC or body mass index among 14,176 participants in Germany and Italy.” (Marked in blue, Page 13, Lines 284-287)

Q8 “Measurement of aging biomarkers” subsection: A substantial part of results derived from molecular biology techniques application; thus, Authors should at least include exact sequences of primers for studied genes and housekeeping genes in the study along with conditions of reactions. It can be provided as supplementary material).

Response: Thanks for your kind suggestions which give us much help to perfect this manuscript. More detailed information on measurement of aging biomarkers and the primers of telomere repeat copy number, mtDNA-CN, and single-copy gene copy number has been supplemented in the Supplementary materials, and sentences were as follows: The real-time polymerase chain reaction (RT-PCR) (QuantStudio™ 7 Flex, AppliedBiosystems Life technologies, USA) was used to measure the telomere repeat copy number, mitochondrial DNA copy number as well as single copy gene copy number in peripheral leukocytes of three times [1]. The forward and reverse telomeric primers were 5′-ACACTAAGGTTTGGGTTTGGGTTTGGGTTTGGGTTAGTGT-3′ and 5′-TGTTAGGTATCCCTATCCCTATCCCTATCCCTATCCCTAACA-3′, respectively [2]. The primers of mtDNA forward and reverse were 5′-TGGCTCCTTTAACCTCTCCA-3′ and 5′-GGTTCGGTTGGTCTCTGCTA-3′, respectively [3]. The forward and reverse primers for β-actin were 5′-ACTCTTCCAGCCTTCCTTCC-3′ and 5′-GGCAGGACTTAGCTTCCACA-3′, respectively [4]. The thermal cycling conditions of RT-PCR were set at 95°C for 3 minutes, followed by 40 cycles of denaturation at 95°C for 3 seconds, annealing and extension at 60°C for 30 seconds. The melting curve analysis of each PCR reaction product was used to confirm the specificity of primers. The average of three measurements from the cycle thresholds (Cts) of telomere and β-actin were used for analysis. Finally, the relative TL or mtDNA-CN was computed by the average of three measurements of telomere repeat copy number or mtDNA-CN divided by the average of three measurements of single copy gene copy number using the 2△△Ct method [3, 5].

The following references were supplemented in the supplementary material reference lists

  1. Cawthon RM. Telomere measurement by quantitative PCR. Nucleic Acids Res 2002; 30: e47.
  2. Cawthon RM. Telomere length measurement by a novel monochrome multiplex quantitative PCR method. Nucleic Acids Res 2009; 37: e21.
  3. Hou J, Yin W, Li P et al. Seasonal modification of the associations of exposure to polycyclic aromatic hydrocarbons or phthalates of cellular aging. Ecotoxicol Environ Saf 2019; 182: 109384.
  4. Pieters N, Janssen BG, Dewitte H et al. Biomolecular Markers within the Core Axis of Aging and Particulate Air Pollution Exposure in the Elderly: A Cross-Sectional Study. Environ Health Perspect 2016; 124: 943-950.
  5. Livak KJ, Schmittgen TD. Analysis of relative gene expression data using real-time quantitative PCR and the 2(-Delta Delta C(T)) Method. Methods 2001; 25: 402-408.

Q9 The Authors may consider moving Figure 1 to the material and methods section.

Response: Thanks for your insightful suggestion. According to your suggestion, Figure 1 has been moved to the Materials and methods section.
